# Children as a Reflection of Transcendence in the Filmography of Andrei Tarkovsky

**Irena Sever Globan** [1,*] **and Marin Pavelić** [2]

1 Department of Communication, Catholic University of Croatia, 10000 Zagreb, Croatia
2 Netokracija d.o.o., 10000 Zagreb, Croatia; marin@netokracija.com
* Correspondence: irena.sever@unicath.hr

**Abstract:** Andrej Tarkovsky is a Russian film author who has indebted the entire world's cinematography with his cinematic style. His (auto)biography and filmography give us a hint that he was a deeply religious man who believed that art should serve to deepen man's spirituality. By watching and analyzing the author's films, we came to the hypothesis that Tarkovsky uses the characters of children to express something supernatural, and therefore, we wanted to explore which narratives and stylistic devices the director uses to give his interpretation of the spiritual and transcendent. Thus, we analyzed nine characters of children that appear in the director's six full-length feature films: Ivan Bondarev (*Ivan's Childhood*), Boriska (*Andrei Rublev*), Aleksej, Ignat and Asafjev (*Mirror*), Marta (*Stalker*), Domenico's son and Angela (*Nostalghia*), and Gossen (*The Sacrifice*). The methods we have used are qualitative content analysis, description, comparison, and synthesis. The characteristics we have noticed in the characters of the children, which could point to the transcendent, are a deep and penetrating gaze, the supernatural powers children use, the mysterious environments they inhabit, the deep influence they have on other characters, asking religious questions, hermit-like loneliness, modest clothes, and allusions to a Christ-like figure.

**Keywords:** Andrei Tarkovsky; transcendence; children; transcendental style; sacred art; cinema





## 1. Introduction

Since it first saw the light of day almost 130 years ago, the medium of film has continued to capture the attention of both the general public, which consumes it with unwavering enthusiasm and escapism, and the academic community, which recognizes film as an art form on textual and aesthetic levels. In 1963, the Second Vatican Council, in its Decree on the Media of Social Communications (Inter mirifica), emphasized the importance of engaging with contemporary film by believers (Second Vatican Council 1963), as film is not merely an entertainment medium but also directs and shapes the thinking of its viewers by satisfying their mental and spiritual needs (Sever Globan 2020, p. 262). It was in the mid-20th century that texts analyzing the relationship between film and theology began to be published relating to both religious and secular films. However, it was only in the 1980s that a new and autonomous scientific approach began to develop, placing the complex and significant relationship between media, religion, and culture at the center of analysis (White 2007). The premise of this interdisciplinary field of research is the recognition that encounters with the sacred do not occur exclusively within consecrated spaces, in explicitly religious symbols and theological texts. Artifacts of popular culture, such as film, can also become sites of transcendent experience. Therefore, we should not perceive cinematic art solely as a mere object of entertainment but rather as a collection of life stories that simultaneously interpret humanity and are interpreted by humanity. These stories have the power to change lives, communicate truth, redeem (Marsh 1997, p. 22; Johnston 2004, pp. 33–34), and inspire our ongoing search for meaning (Ferlita 1982, p. 117).

Art, as the bearer of the beautiful, has its source in the relationship with the sacred and can communicate the spark of divine presence in the world. Thus, we can assume that

film, if in service to art, can become via *pulchritudinis* and awaken the *homo religiosus* within individuals (Marty 1997, pp. 134–35). One of the most influential film critics and theorists, Andre Bazin, also argued that the aesthetic dimension of film can contain a sacramental dimension. Through its cinematography, the camera, as a meditative eye, can bear witness to the miracles of God's creation (Bazin 1992). While it is impossible to depict God on the screen, in some cases, it is possible to recognize Him as a force acting within the film's protagonists, motivating them to actions that lead to life-changing experiences. When a film suggests that man is not merely a byproduct of chance but in him resides something beyond human, "something that needs to be awakened to vibrant life by a divine touch" (Škvorc 1982, p. 300), the director has stepped onto sacred ground, even if he may not be consciously aware of it. In this sense, it is not necessary for a film to use religious or explicitly theological language to be characterized as a religious film that evokes the sacred, nor is it necessary for the director to be religious himself (May 1982, p. 28). A theological film, in fact, would be one that questions the meaning of man's existence and the divine grace that can be clothed in a secular setting (Greeley 1988, p. 97). Along these lines, in a recent interview with an Italian newspaper, the Italian director and self-proclaimed atheist Marco Bellocchio, ahead of the premiere of his new film *Kidnapped*, based on a true story of a Jewish boy abducted by the Papal State and converted to Catholicism, referred to Carl T. Dreyer's film *Ordet*, which evoked sacred feelings within him. Bellocchio stated, "(...) more than many theologians, it is the great artists, painters, writers who have brought people closer to religion. Those are the great carriers of a possible faith. The way of beauty" (Mallamo 2023).

Once, one of the most significant Russian (and worldwide) directors of the 20th century, Andrei Tarkovsky, attempted to explain what being a filmmaker meant to him, drawing a parallel between the experience of filmmaking and the transcendent: "(...) my films are not a personal expression but a prayer. When I make a film, it's like a holy day. As if I were lighting a candle in front of an icon, or placing a bouquet of flowers before it" (Gianvito 2006, p. 175). Tarkovsky took the art of filmmaking extremely seriously, considering a film to be more than just a step in one's career; it becomes an act that will impact a person's entire life. Childhood is also an important theme that often serves as a leitmotif in many artistic expressions, including those of Tarkovsky. The sculptor Constantin Brancusi claimed that an artist must possess childlike qualities to be creative. Many artists view childlike nature as a crucial element of art (Robinson 2006, p. 294). Tarkovsky echoes this sentiment when he states that a poet is someone who possesses the psychology and imagination of a child (Gianvito 2006, p. 137). Andrei Tarkovsky is a globally renowned filmmaker who has been extensively studied and written about. However, this paper aims to offer originality by analyzing something that has not been systematically explored in relation to his filmography: the depiction of child characters as a possible reflection of the transcendent. We believe that Tarkovsky expresses the transcendent through the child characters in his six feature films (*Ivan's Childhood*, *Andrei Rublev*, *Mirror*, *Stalker*, *Nostalghia*, and *The Sacrifice*), which forms the hypothesis of this article. The goal is to identify and analyze the narrative and stylistic devices the director employs to portray the spiritual and transcendent. In the analysis of the selected films and child characters, we employ qualitative content analysis, description, comparison, and synthesis methods to understand whether and how the child characters reflect the transcendent and spiritual reality.

## 2. The Transcendent in (Cinematic) Art

Paul Schrader believes that art can provide glimpses of the sacred and transcendent, and this is achieved through a specific style of depicting reality, which he calls the transcendental style. Anthropologists have noted that artists from different cultures have discovered similar ways of expressing equivalent spiritual emotions and experiences at the turn of the 19th to the 20th century. In the realm of film, filmmakers from different eras and regions have established a consensus around the transcendental style. The style itself is not inherently transcendental or religious, but it serves as a path toward the transcendent. The

specific aspect being transcended varies in each individual case, but the goal and method, in their purest form, remain the same (Schrader 1988, p. 3).

The term "transcendent" originates from the Latin word transcendere, which translates to "to surpass". Its specific meaning depends on the areas in which it is applied. In the context of epistemology, it denotes the independence of the cognitive object from cognition and the cognitive subject, while in metaphysics, it refers to the transcendence of any sensory experience and existence beyond the visible world. In theological terms, the transcendent pertains to God as the one who exists completely outside the existing sensory world. In the context of science, the transcendent encompasses everything that surpasses the possibility of scientific proof and points to a reality that humans cannot touch (Ivančić 1980, pp. 243–44). Martin Heidegger defines the transcendent as man's distancing from the world of objects in his own being to experience the world in its entirety (Heidegger 1995). Karl Jaspers interprets the concept as that which eludes theoretical knowledge and reveals itself only through concrete decision-making and can be discovered through comparison (Jaspers 1971). Mircea Eliade believes that man becomes aware of the sacred or transcendent because it manifests as something "wholly different from the profane" (Eliade 1987, p. 11); it is a reality that does not belong to this world and an appropriate expression for it would be hierophany. Hierophany is a mysterious act, a manifestation of the sacred that can occur through various objects in different religions, such as stones or wood, but it can also be highly elevated, as in Christianity, where God incarnated in Jesus Christ (Eliade 1987). Rudolf Otto uses the term numinous to denote the feeling that arises when one encounters the transcendent. He describes the numinous as a moment of "terror and awe", which is not merely an innate fear but a glimpse of the mysterious and the transcending of the natural realm, which also reveals itself in art (Otto 1923). Offenbacher states that the transcendent "designates this wholly other reality which one can feel, without being able to give it a clear conceptual expression. (...) In art, man's struggle can be emphasized discerning the transcendence that occurs at the boundary of finitude" (Offenbacher 1985, p. 83).

By using the term transcendent art, what is meant is that religion and art are connected. Gerardus van der Leeuwa claims that "Art can be religious or can appear to be religious; but it can be neither Mohammedan nor Buddhist nor Christian. There is no Christian art, any more than there is a Christian science. There is only art which has stood before the Holy" (cited according to Schrader 1988, p. 7). The purpose of transcendent art is to express the sacred (Schrader 1988). Pope John Paul II was following this line of reasoning when he wrote in his *Letter to Artists* in 1999: "None can sense more deeply than you artists, ingenious creators of beauty that you are, something of the pathos with which God at the dawn of creation looked upon the work of his hands" (John Paul II 1999, p. 3). Next, he states that God has called man into existence and entrusted him with the task to share in his creative work: "Through his 'artistic creativity' man appears more than ever 'in the image of God', and he accomplishes this task above all in shaping the wondrous 'material' of his own humanity and then exercising creative dominion over the universe which surrounds him. With loving regard, the divine Artist passes on to the human artist a spark of his own surpassing wisdom, calling him to share in his creative power" (John Paul II 1999, p. 5). Tarkovsky would probably agree with this notion as he expresses a similar sentiment when he claims: "(...) we are created in the image of God and share this creativity with God" (Gianvito 2006, p. 160), or: "Art affirms all that is best in man—hope, faith, love, beauty, prayer... What he dreams of and what he hopes for... When someone who doesn't know how to swim is thrown into the water, instinct tells his body what movements will save him. The artist, too, is driven by a kind of instinct, and his work furthers man's search for what is eternal, transcendent, divine—often in spite of the sinfulness of the poet himself" (Tarkovsky 1986, p. 239).

Transcendental style in the seventh art has been utilized by various artists from different cultures to express the sacred. As emphasized by the author of the book *Transcendental Style in Film: Ozu, Bresson, Dreyer*, Paul Schrader, a transcendental style would be a way

of filming that seeks to absolutize the mystery of existence, to depict what lies beyond the physical and everyday perception of the Other. It is fundamentally a style, meaning it can be isolated, analyzed, and defined. Such an expression aims at the invisible and inexpressible, yet it is not invisible and inexpressible itself. It employs various camera angles, dialogues, and editing techniques to bring forth the sacred (Schrader 1988, pp. 3–4). What expresses the sacred and the transcendent in film, therefore, is the film's style and techniques that constitute its aesthetics, such as slow camera movements, long takes, the use of silence, specific editing, the utilization of close-ups, facial expressions, and more. Let us recall that the French director Robert Bresson believed that if we gaze at the expressive faces of actors long enough, we will see the mystery of God's image.

The goal of the transcendental style is to heighten the mystery of existence while avoiding conventional interpretations of reality. Films that employ conventional interpretations of reality, for an artist seeking to express the transcendent, become emotional and rational constructs that serve to explain and mitigate the transcendent. Robert Bresson refers to these constructs as "screen traces" to help the audience better understand the plot, acting, characterization, camera work, music, dialogue, and editing. However, in the transcendental style, these elements lack expressiveness, meaning they do not convey culture or personality but are reduced to a state of inertia. An artist who aims to express the transcendent rearranges reality by removing elements that primarily express human experience, thereby observing reality without resorting to conventional interpretations. Thus, the transcendental style transforms the experience into a repetitive ritual that can become transcendent with each repetition (Schrader 1988, p. 11).

In the article *Film as a Transcendental Experience*, Elmer Offenbacher argues that film can evoke the transcendent because the seventh art can convey an "illusion of sacred reality" to the viewer (Offenbacher 1985, p. 85). In this way, film recognizes a mysterious world that could maintain a relationship with the real world, whereby this relationship is familiar to us. Through this process, film is humanized, and the image serves as a means of our communication with God (Offenbacher 1985). External factors such as darkness in the cinema hall or the silent attention of the audience are not sufficient to replace deeper characteristics such as a slow rhythm, meditation, and discernment leading to the conclusion in order that the viewer can have a transcendent experience while watching the film. Offenbacher asserts that among all the possible techniques available to a director, the close-up shot is the one closest to the tool of transcendence, and the first director to use the close-up shot for a transcendent purpose was Carl T. Dreyer. Offenbacher believes that the purpose of transcendental style is potentially the practice of empathy. When the image pauses, the viewer can delve deeper into it, approaching the silence in which religion and art intertwine and possibly transcend the realm of art itself (Offenbacher 1985, p. 101).

## 3. The Film Style of Andrei Tarkovsky

Andrei Arsenyevich Tarkovsky was a Russian director, screenwriter, and actor who was born in 1932 in Zavrazhye and died in 1986 in Paris. He gained worldwide fame with his feature film *Ivan's Childhood* (Tarkovski 1962), after which he directed seven more feature-length films that have become renowned in the history of cinematic art. Tarkovsky never hid his religious beliefs, and religious themes were at the core of his films. He was fascinated by questions of faith, purity, integrity, goodness, evil, doubt, suffering, and sacrifice (Robinson 2006, pp. 261–62). Tarkovsky was a Christian of Orthodox faith and a practicing believer, and in accordance with this, he stated: "Faith is the only thing that can save man, that's my deepest conviction" (Gianvito 2006, p. 186), and: "For my aim is to show life, to render an image, the tragic, dramatic image of the soul of modern man. In conclusion, can you imagine such a film being directed by a nonbeliever? I can't" (Gianvito 2006, p. 179). The influence of the Orthodox Church is evident in his films through recurring iconographic depictions of candles, icons, fire, crosses, and religious rituals, as well as references to the monastic way of life and devotion to the Virgin Mary (Robinson 2006, p. 276). He considered his films a kind of prayer: "I'm a man to whom God

gave the possibility of being a poet, meaning, of praying in another manner than the one used by the faithful in a cathedral" (Gianvito 2006, p. 166). In conversation with the writer Laurence Cossé, when asked if his faith in God aligns with the faith he portrays in his films, Tarkovsky responds: "Art is the capacity to create, it's the reflection, the mirror-image, of the Creator's gesture. We artists only repeat, only imitate that gesture. Art is one of those precious moments in which we resemble the Creator. That is why I have never believed in art which would be independent of the supreme creator, I don't believe in art without God. The raison d'etre of art is a prayer, it's my prayer. If this prayer, if my films can bring people to God, so much the better. My life would then take on its sense, the essential sense of 'serving'" (Gianvito 2006, pp. 170–71).

Film critic Dina Pokrajac highlights Tarkovsky as a director who nurtures the religious element of art, describing art as a "hieroglyph of absolute truth" and a "detector of the absolute" (Pokrajac 2019, p. 148). Through art, infinity becomes tangible, and the idea that faith and creativity are one lies at the core of Tarkovsky's poetics. Art employs symbols to reveal the purpose of human existence, and for Tarkovsky, that purpose is to lift the spirit up, achieve refinement, and attain harmony with others and the universe. Creation, for Tarkovsky, is a difficult and lengthy act leading to redemption, but the artist is a figure of tragedy. The artist should not be proud because his role is to serve. Pokrajac concludes that Tarkovsky is in search of a mystical participation between the world and man. He condemns the gap between the world and man caused by science, monotheism, and mechanistic labor organizations (Pokrajac 2019).

Tarkovsky created a unique directing style, becoming one of the most significant representatives of auteur film. His films are poetic, artistic, and not intended for entertainment. His body of work is characterized by a rich diversity of stylistic approaches, ranging from features reminiscent of classic Soviet film in his early works to complex symbolism, pessimism, and elements of Orthodox mysticism. As an advocate of authorial cinematography, Tarkovsky wrote the following in his autobiographical book *Sculpting In Time*: "He starts to be an artist at the moment when, in his mind or even on film, his own distinctive system of images starts to take shape—his own pattern of thoughts about the external world—and the audience are invited to judge it, to share with the director in his most precious and secret dreams. Only when his personal viewpoint is brought in, when he becomes a kind of philosopher, does he emerge as an artist, and cinema—as an art." (Tarkovsky 1986, p. 60).

For Tarkovsky, there were two ways to connect shots into a whole. The traditional method was linear, where the plot follows a logical sequence, and sequences are connected by understanding the order or focusing on the characters. Tarkovsky considered this to be a simplified interpretation of reality, which is far more complex. Consequently, he devised his idea of the logic of poetry. This concept does not refer to a literary genre but rather to a way of thinking that emphasizes associative material, resulting in a non-chronological presentation of the plot. The director follows the logic of human thought when making the film. His aim is to evoke reactions and emotions in viewers rather than provide a ready-made idea, which is why the audience plays a crucial role in shaping the purpose of the film. Viewers have significant freedom in interpretation, consequently leading to different understandings (Dragičević 2020).

Robinson points out that in Tarkovsky's films, we can recognize two different tendencies when setting the mise-en-scène. One strives for the transcendent and universal, while the other is characterized by a mise-en-scène that is real and messy. He emphasizes that at the core of Tarkovsky's films lies the tension between the symbolic, religious, and iconographic and real, plastic, and immanent. According to Tarkovsky, the mise-en-scène should not be schematic or clichéd but should surprise us with authenticity (Robinson 2006, p. 176). An essential element in Tarkovsky's filmmaking is the transition from narrative digression to "dead time" (Schrader 2018, p. 9). There is a fundamental difference in the purpose for which time-consuming long takes are used. They primarily serve to create an atmosphere, but they also engage the audience. What Bresson and Ozu aspired to, Tarkovsky accomplished. He was not a complete adherent of slow cinema, but he made

it fashionable (Schrader 2018, pp. 9–10). Slow cinema is a term that defines a branch of art cinema characterized by minimalist storytelling and cinematography, long takes, and extended film duration. The key features of slow cinema include long takes, wide shots (providing more information for the audience to focus on), static shots, minimal use of different shots, offset edits (cuts that are intentionally too early or too late), favoring visuals over dialogue (dialogue is unconventional), highly selective use of music or no music at all (sound serves a diegetic function), pronounced sound effects (filling the empty space without music and dialogue), visual monotony (shots tend toward symmetry without specific visual information), recurring shot compositions, doubling (repeating the same information), and omission of acting (characters resemble figures within the environment). A film does not necessarily need to have all the listed characteristics to be considered slow cinema; Schrader refers to them as a "buffet" of technical possibilities, some of which can be in opposition to each other (Schrader 2018, pp. 11–16).

## 4. Children and Childhood in the Movies by Andrei Tarkovsky

In an interview related to the topic of understanding his movies, Andrei Tarkovsky stated: "You have to be a child—incidentally children understand my pictures very well, and I haven't met a single serious critic who could stand knee-high to those children. We think that art demands special knowledge; we demand some higher meaning from an author, but the work must act directly on our hearts or it has no meaning at all" (Gianvito 2006, p. 67). Clinical psychologist Jordan B. Peterson shares this opinion seeing children as "small and young and, in some ways, they do not know anything, because they have very little personal experience. But they are also very ancient creatures, in another manner, and by no means stupid or inattentive. The fact that they are gripped by fairy tales and stories like Pinocchio is an indication of just how much depth children perceive in those stories, even if you, as an adult observer, do not notice it anymore" (Peterson 2021, p. 147). Children and childhood are popular cinematic topics from the beginning of cinema, often conveying adult's (director's) perspectives of certain issues regarding society (Wibawa 2010). At the end of the 19th century, the Lumière brothers had made the first film about children, "and soon thereafter virtually every film culture grasped the new possibilities of capturing children's cuteness and mischief and pathos" (Cardullo 2001, p. 295). Children and childhood are also a leitmotif in many of Andrei Tarkovsky's films. They serve as carriers of the narrative, possessing unique knowledge, and are subjects of supernatural abilities.

Important to emphasize about Tarkovsky's films is that their vast majority belong to the period of Soviet Thaw of the 1950s and 1960s in the 20th century. The common characteristic of the Soviet Thaw films is the representation of children and youth born in the turbulent times of great socio-political uncertainties and cultural changes, together with the generation gap between children and society.

And, "while adults destroy their dreams, children try to realize them. A child of the Thaw is not a helpless creature lost in the city; he is a personality who tries to find his place in the huge world" (Ianushko 2021). Alena Ianushko mentions and analyzes in her article about portraying children in films of the Soviet Thaw period some of the most significant films of that period, like *Splendid Days* (1960) by Georgy Danelia and Igor Talankin, *Destiny of a Man* (1959) by Sergej Bondarčuk, *The Boy and the Dove* (1961) by Andrej Končalovski, *Welcome, or No Trespassing* (1964) by Elem Klimov, *Ilych's Gate* (1965) by Marlen Khutsiev, *Goodbye Boys* (1964) by Mikhail Kalik, etc. She claims that "the idea of a child's consciousness is an essential feature of the Thaw" (Ianushko 2021), as well as the fact that a child's character from soviet films of that age is shown as the one having the right to say what it means and often has a deeper understanding of the world than adults who fall short of advice. Young hero in soviet films is connected with archetypes of Home and Road: "The archetype of the house symbolizes stability, whereas the concept of the road represents movement and development" (Ianushko 2021). Another theme appearing in Thaw cinema is a young character with a war-torn childhood, as we shall see in *Ivan's Childhood*. Poetic

representation of children in this period of soviet films is characterized by showing their world through dreams of flying, traveling, and reaching the sea or sky, as a sort of escapism from bitter reality. Thus, a new film hero of Cinema of the Soviet Thaw is "a little man who is trying to finds his place in the world" (Ianushko 2021).

Ianushko compares the soviet film with French cinematography, especially the French New Wave, which covers the topic of youth freedom and awareness in decision-making. Just remember child characters in films like *400 Blows* (1959) by François Truffaut, *Zazie in the Metro* (1960) by Louis Malle, or the short film *The Red Balloon* (1957) by Albert Lamorisse. Bert Cardullo agrees with Ianushko when he states that "In the vein of juvenile performance—with professional child actors as well as non-professionals or 'non-actors'— no movie culture has done better than France, however" (Cardullo 2001, p. 295). The author reflects here on somewhat more modern films like *Ponette* (1996) by Jacques Doillon or *It All Starts Today* (1999) by Tavernier about "preschool children living amidst Zolaesque conditions in contemporary northern France" (Cardullo 2001, p. 295). Italian cinematography stands out too in film depiction of children as the central focus of the narrative, especially in the period of Italian neorealism. In it, children were depicted as innocent and pitiful victims of unfavorable societal forces. At the same time, children were a vessel of hope and renewal. Let us remember films like *The Children Are Watching Us* (1943), *Bicycle Thief* (1948) and *Shoeshine* (1947) by Vittorio de Sica, *Germany, and Year Zero* (1947) and *Rome, Open City* (1944) by Roberto Rosselini (Cardullo 2001, 2015; Hipkins and Pitt 2014).

Andrei Tarkovsky's last five films—*Solaris*, *Mirror*, *Stalker*, *Nostalghia*, and The *Sacrifice*—end with an affirmation of family, parental relationships, and childhood. In films such as *Ivan's Childhood*, *Solaris*, and *Mirror*, Tarkovsky explores the world of childhood, which man remembers through memories, wishes, and regrets. He achieves this by using mnemonic film techniques. Like many other artists, this Russian director also explores his own childhood and psychology through his films to re-experience the pain he underwent. His films serve as fairy tales; he uses mechanisms in them in which the subconscious is not censored, while anxiety is strongly expressed. It portrays childhood as extremely mysterious; everything in it is unusual and foreign but never threatening (Robinson 2006, p. 295). Children in Andrei Tarkovsky's films are not exposed to excessive sentimentalization and do not have idealized and innocent childhood, as is often the case in Hollywood films (Wibawa 2010; Chaturvedi and Verma 2022), with of course some bright examples of wise and "otherworldly" children in films such as *The Night of the Hunter* (1955) by Charles Laughton, *The Sixth Sense* (1999) by M. Night Shyamalan, *The Tree of Life* (2011) by Terence Mallick, *Beasts of the Southern Wild* (2012) by Benh Zeitlin, *The Book Thief* (2013) by Brian Percival, *Heaven is for Real* (2014) by Randall Wallace, etc.

In Tarkovsky, children are closer to the magical and supernatural world than adults. Thus, Alexei from *Mirror* has premonitions, Boriska from *Andrei Rublev* has supernatural powers, and in *Nostalghia*, the boy asks the following question with all seriousness: "Is it the end of the world, Dad?", the boy from *The Sacrifice* lies under a tree like Buddha and utters the divine words of creation: "In the beginning was the Word". Robinson finds a parallel in Fellini's way of depicting reality and childhood when he writes: "Andrei Tarkovsky's cinematic depiction of childhood has affinities with that of Federico Fellini: the bed, the furniture, the house and its environs are seen as magical spaces, where the individual was first formed, where her/his first encounters with a larger world took place In films by both Federico Fellini and Tarkovsky sheets and curtains billow mysteriously in the wind; the house is presided over by a Madonna; words are not spoken, nor rememberedchildhood is experienced primarily, sensually, physically, viscerally, and before language" (Robinson 2006, p. 295).

By analyzing Andrei Tarkovsky's feature-length films in which children's characters appear, we want to explore the relationship between the transcendent and childhood, and therefore, the aim of the paper is to identify which narrative and stylistic means the director uses to give his interpretation of the spiritual and transcendent. The main hypothesis of the research is the following: Andrei Tarkovsky, with the help of children's characters, tries

to express the transcendent by depicting them in unusual situations and with supernatural abilities, most often alluding to the Christian religious tradition.

The main research questions are the following:

1. What stylistic devices does Tarkovsky use in the depiction of children, and can they be a reflection of the transcendent?
2. What are the psychological characteristics of children depicted in Tarkovsky's films?
3. How do the children's characters behave?
4. What kind of environment are the children's characters in?
5. What do the children's characters look like?

To answer the above questions, the selected film scenes are analyzed using the method of qualitative content analysis, "a procedure of studying and analyzing verbal or non-verbal material in order to discern its characteristics and messages" (Lamza Posavec 2011, p. 105). Also, to better understand the children's characters, we use the semiotic analysis of film through the codes of denotation and connotation (Barthes 1986, p. 91), as well as the analysis and synthesis of the narrative.

The films selected for the analysis are those in which children's characters appear, either main or supporting characters essential to the plot. Therefore, we analyze the following films and children's characters: Ivan Bondarev from *Ivan's Childhood*; Boriska from *Andrei Rublev*, Alexei, Asafyev and Ignat from *Mirror*; Marta from *Stalker*; Domenico's son and Angela from *Nostalghia*, and Gossen from *The Sacrifice*. For each film, we provide a summary and then focus on the analysis of scenes in which we have identified elements that could point to the transcendent, related to the depiction of children.

### 4.1. Ivan in Ivan's Childhood (Tarkovski 1962)

*Ivan's Childhood* is the first feature-length film by Andrei Tarkovsky, centered around Ivan Bondarev, a twelve-year-old orphan and scout during World War II. The story begins with Ivan's dream and waking up, after which the boy traverses the war-ravaged landscape until he reaches a river, which he swims across. On the other bank, he is captured by Soviet soldiers and taken to the young Lieutenant Galtsev. Through a series of dreams and conversations with different characters, it is revealed that Ivan lost his entire family during the war: his mother and sister were killed by the Germans, and very likely his father as well. This filled the boy with hatred, so he adamantly seeks revenge. In the end, Galtsev discovers a document attesting that Ivan was captured and hanged. The film is interspersed with sequences of Ivan's dreams and historical footage from World War II. The plot of the film is subordinated to the psychological experience of war, which is indirectly depicted by emphasizing the consequences left behind by the war. Robinson defines the film as "a superb depiction of the loss of innocence in childhood, how youth is robbed of childhood, how the sins of the fathers utterly wreck a young soul" (Robinson 2006, p. 317). According to Ianushko, the plot of this Tarkovsky film is based on the "archetypes of road and home. The road represents a path of the young characters, and the concept of home shows their ultimate goal. The halfruined house shown in the last episode becomes the place where Ivan's memories of unlived childhood come to life. 'I am my own master', Ivan says to adults who try to control his life" (Ianushko 2021).

Tarkovsky creates a contrast by juxtaposing the sunlit childhood and peaceful dreams with the bleak battlefield where Ivan is neither a child nor an adult. There is no transition between waking and dreaming, only a stark cut that opposes tranquility and unrest. These two worlds exist in different times but in the same space. Tarkovsky himself describes it as the coexistence of beauty and ugliness in one. The changing landscape reflects the destructive power of war and Ivan's loss of innocence (Green 1993, pp. 26–27). Actually, in many of his films, Tarkovsky depicts how war destroys the natural order of things, taking with it all positive images (Ianushko 2021).

According to Antoine de Baecque, Ivan is referred to as a victim, a saint, and a martyr. He is seen as both a monster and an innocent child seeking revenge for his mother's death (according to Robinson 2006, pp. 330–31). His childhood has passed, but he has only

partially matured. He has developed a commanding tone of speech, carries himself like an officer, is willing to make sacrifices, and feels the futility of his own existence. The limits of his inner toughness are revealed when he asks Galtsev for a knife or when he must be carried to bed due to exhaustion (Green 1993, p. 29). For Ivan, war and violence are the only reality, while dreaming is the only respite and comfort. Turovskaya interprets his dreams not as memories but as images of freedom, an imaginative game, and vague pantheistic visions of a normal life that is peaceful and filled with happiness (Turovskaya 1990, p. 6). The film contains four dream sequences, which serve not only to express Ivan's suffering and that of the Russian people but also to demonstrate how, in the early stages of his career, Tarkovsky divided the world into what is internal and external (Martin 2005, p. 68).

To illustrate Ivan's connection with the transcendent, we analyze the first and the second dream sequence. The first dream also serves as the film's beginning and starts with a close-up shot of Ivan standing behind a tree, the branches of which are connected by cobwebs. Ivan then walks away, and the camera moves up toward the top of the tree. In the background, Ivan returns, but now we see him in full. After we see a goat, Ivan runs through the forest and notices a butterfly. At that moment, Ivan begins to levitate and descends to the beach, where he finds his mother holding a bucket of water. He tells her, "Mom, there's a cuckoo over there", and she smiles at him. However, the camera tilts and quickly approaches her worried face while we hear gunshots and Ivan's loud calling for his mother in the background. According to some interpretations and beliefs, the cuckoo bird would be a messenger from "the other world" or the soul of the deceased. Disguised as a cuckoo, the soul seems to fly down to Earth to converse with relatives, which makes sense since Ivan's mother is dead. During this dream sequence, Ivan displays a transcendent ability: levitation. While levitating, the blue-eyed and blond-haired Ivan appears angelic, showing signs of happiness and exhilaration on his face. The phenomenon of levitation is written about in the New Testament and Torah, and supposedly, many saints like Francis of Assisi, Thomas Aquinas, and Ignatius Loyola have experienced this paranormal ability. Now, Ivan joins them, at least in his dreams.

The second dream begins with a detail of Ivan's palm from which the water is dripping, and by moving the camera to the left and up, Tarkovsky reveals that we are at the bottom of the well. At the top, leaning against the well, are Ivan and Mother, who explains to him that despite the sunny day, it is still possible to see a star in the deep well. Ivan reaches towards the well and stirs the water with his touch, which reveals that the camera is filming underwater. Ivan is now at the bottom of the well and tries to catch the star with his hands while Mother lifts a bucket filled with water, but the bucket starts to fall suddenly, and Ivan tries to avoid it. The mother lies dead on the ground, and the water, raised by the blow of the bucket on the bottom of the well, falls on her. This sequence points to where the metaphorical Ivan is. Ivan's life is on the border of day and night, more precisely, between the spiritual world inhabited by the person most important to him and the real world at war. This division is visible in the sentence in which Mother addresses Ivan: "For us it is day, but for the star it is night". The mention of day and night can signify life and death on a connotative level. Since both the mother and Ivan are dead at the end of the film and thus partakers of the afterlife, this can also symbolize the thought that they are in the light, where there is day, that is, with God, while the rest of the people on Earth are still in the night like the stars. The whole sequence is shrouded in a sense of mystery as we never find out why the star is at the bottom of the well; instead, this fact is taken as a normal occurrence. Additionally, for an unknown reason, Ivan tries to catch the star with his hands. The star is a symbol of the spirit and the struggle between spiritual and material forces or light and darkness due to its celestial characteristics. In their light, the manifestation of supernatural powers that govern the destiny of humanity was perceived. Guided by this thought, we can assume that Ivan wants to grasp the power through which he will govern the destiny of mankind and then seek revenge for his family's death. To emphasize the transcendent feeling of the place where Ivan is, Tarkovsky plays with the viewer's

perception of space using camera shots. The first shot begins with a sleeping Ivan, whose bed is in the well.

Next follows a conversation between Ivan and Mother, presented in a way that gives the impression that we are watching their reflection in the well. However, as small pebbles fall into the well, it is revealed that the camera is recording from a frog's perspective, positioned underwater. Thus, Tarkovsky aims to disrupt the viewer's naturalistic expectations by placing the camera underwater.

The transcendence of Ivan Bondarev can be recognized in his duality between the world of reality and the world of dreams or, on a metaphorical level, between life and death. The events in the dream world act as a "voice of God" that calls him to take selfless actions to stop the bloodshed and defend his country, following the example of the saint and martyr Joan of Arc. The dream sequences are portrayed as an idyllic place where there is no suffering, and Ivan is happy, living in harmony with God and nature, no longer constrained by the laws of physics. These dream sequences take place in a forest and on a beach, bathed in bright sunlight and filled with children's laughter. Tarkovsky used these dream sequences to depict what is referred to in Christianity as "heaven", symbolizing the transcendence and spiritual unity with God. Hence, his name, which translates to "God is gracious", is not coincidental. And if a saint is a bridge between heaven and Earth, and Ivan is depicted as someone who connects heaven and Earth, then he, in a certain sense, becomes a saint himself, but also a martyr as he dies a martyr's death, killed by military enemies.

### 4.2. Boriska in Andrei Rublev ([Tarkovski 1966](#))

*Andrei Rublev* is Andrei Tarkovsky's second feature-length film, centered around the eponymous Russian medieval icon painter. The film consists of eight episodes, as well as a prologue and an epilogue. In the first seven episodes, we follow the life of Andrei Rublev from 1400 to 1423, during which he becomes an assistant to the renowned icon painter Theophanes the Greek. He encounters pagans and witnesses their rituals, loses his fellow workers who are killed by the Grand Duke, and survives a cruel attack by the Tatars, during which he prevents a rape by killing one of them. As a result of this act, he renounces icon painting and takes a vow of silence. In the final, eighth episode, The Bell, a boy and orphan named Boriska appears, who will have a profound impact on Rublev. The soldiers of the Grand Duke are looking for Boriska's father, but Boriska tells them that his entire family was killed by the plague. He convinces them that only he now possesses the secret of bell casting, and the soldiers believe him, appointing him responsible for casting the church bell. Boriska is a fourteen-year-old young man with blue hair and blue eyes, reminiscent of an angel just like Ivan, dressed in typical Russian peasant clothing, attempting to make the bell while being secretly observed by Rublev, who becomes a secondary character in this episode. The process of bell casting, led by Boriska, is shrouded in smoke and the light produced by the glowing metal, giving an impression of something supernatural being created.

Robinson refers to the entire bell-casting process as a religious vision that resembles the appearance of a painted Baroque ceiling in a church ([Robinson 2006](#), p. 367). After successfully casting the bell, it needs to be solemnly tested before its patron, the Grand Duke. Boriska and his workers feel great stress, knowing that their lives are at stake if the bell is not made well. Tarkovsky builds tension as the foreman swings the tongue of the bell until we finally hear the resounding sound that only a perfectly crafted bell can produce. After the ceremony, Rublev finds the exhausted Boriska lying in the mud, crying. He lifts Boriska up and holds him in a manner reminiscent of Michelangelo's Pietà—an iconographic motif featuring the figure of the Virgin Mary holding the lifeless body of Jesus in her lap after it was taken down from the cross. This draws a parallel between Boriska and the Christ-like figure.

At that moment, we learn a crucial piece of information: Boriska confesses to Rublev that he had never known the secret of bell casting. This means that Boriska had been guided

by intuition and faith in God all along. Rublev is deeply moved by Boriska's act and faith, and he decides to break his vow of silence that lasted for sixteen years. He tells Boriska that he will continue casting bells while Rublev will paint icons, inviting him to join him on this path. Boriska's faith, reliance on God, and creative intuition awaken in Rublev the desire to create art again and restore his faith in humanity, which he lost when he decided to commit evil to prevent evil, specifically killing a Tatar to prevent a rape. The encounter between Boriska and Rublev is perhaps the most significant moment in the film because it expresses an affirmation of human life and faith in goodness.

Undoubtedly, Boriska is an artist of strong will and faith in his abilities and God. His name itself means "one who fights for glory". His pace and voice exude self-confidence, even though he is aware that he does not possess the secret of bell casting. In his essence, he embodies the role of a holy fool, i.e., someone who deviates from expected societal behavior while displaying supernatural abilities. Tarkovsky further expresses his transcendence by appointing him the leader of casting the new bell, placing him in a situation where he must be a shepherd to his "flock" of workers and assumes the responsibility of someone who will bring joy to his people through the casting of an aesthetically pleasing bell. Visually, Tarkovsky portrays his transcendence in the scene of casting the bell itself. The camera continuously follows him, emphasizing his significance as the master of his mysterious "kingdom" filled with smoke and intense light, creating a contrast like a chiaroscuro painting technique and evoking a sense of something majestic and transcendent—the light appearing behind Boriska's head resembles an aura and represents the power of knowledge. This strong light visually symbolizes something divine and supernatural since in the Bible, as well as in Christian iconography and visual arts, when describing or depicting God, the metaphor of powerful white light, fire, or the Sun is often employed.

### 4.3. Alexei, Ignat, and Asafyev in Mirror (Tarkovski 1975)

*Mirror* is Andrei Tarkovsky's fourth feature film, exploring the thoughts, feelings, and memories of the main character, Aleksei, a forty-year-old man who reflects on his childhood and youth, particularly memories of his mother Maria, and the traumas of war and his parents' divorce. Adult Aleksei is visually glimpsed only briefly, but his voice often accompanies the scenes unfolding before us, which blend pre-war, wartime, and post-war periods. *Mirror* is a ciné-poem filled with metaphors, allusions, lyricism, abstract imagery, various voices, motifs, and symbols. The film employs autobiographical elements, readings of poetry by Arseny Tarkovsky, dreams, flashbacks to the past, newsreels, and mnemonic techniques. Robinson defines the movie as film poetry (Robinson 2006, p. 401). Green believes that immortality is in the center of *Mirror*, and Tarkovsky expresses it thorough the topic of childhood (Green 1993, p. 82). The film is filled with narratives whose trajectory is distorted, and the journey that Tarkovsky takes the viewer on is not material but metaphysical (McFadden 2012, p. 51). *Mirror* represents an invocation of childhood and its mystery and pain. Tarkovsky omits depictions of school attendance and friendships, which results in a somewhat fragmented portrayal of childhood, focusing only on specific parts. As a result, it conveys a sense of loneliness, particularly emphasized by the absence of the parents. Few films about children and childhood possess the originality of *Mirror*, and it avoids sentimentality and self-indulgence. Instead, it is self-reflective and introspective, achieving a universal transcendence (Robinson 2006, p. 433). Tarkovsky, through the child's perspective of the world, attempted to convey what Ernst Bloch, a German philosopher, described as: "something we glimpse in childhood, but where no one has ever been–not merely a place where one belongs, but language and memory, time present and past, and perhaps even time future" (cited according to Green 1993, p. 92). *Mirror* is a film dedicated to the reflection on memory, and its complexity is woven from the imagination of two boys, Ignat and Aleksei.

*Mirror* is a film that features the most child characters. Twelve-year-old Aleksei and Ignat, apart from being related by blood, are also connected by the fact that they are both portrayed by the same actor, Ignat Daniltsev. Tarkovsky attempts to convey the

mysteriousness of the child's world in a sequence that shows Aleksei and his mother visiting their neighbor, Mrs. Solovyova. They leave Aleksei alone in the room. When he looks at his reflection in the mirror, the camera slowly zooms in on his reflection, transforming into a close-up shot of Aleksei's face in the mirror. Tarkovsky then turns the camera back to Aleksei.

The scene is accompanied by classical music that evokes a sublime feeling. Daniel O. Jones believes that in this scene, Aleksei is not looking at himself but rather at his unborn son (Jones 2007, p. 159). Tarkovsky's message is that one will become the other, which is literally expressed by having them portrayed by the same actor. To convey this idea and elevate the scene further, Tarkovsky employs a slow zoom, resulting in a close-up of Aleksei, through which the director attempts to express the supernatural. This shot allows us to fully immerse ourselves in Aleksei's emotional state, who, unlike the emotionally charged Ivan Bondarev and Boriska, remains completely composed. He does not display positive or negative emotions but remains consistently serious, with a deep and focused gaze that gives the impression that Aleksei knows more. These are characteristics shared by most children in Tarkovsky's upcoming films.

During the sequence of the visit, Aleksei notices that there is someone else in the house. In the mirror, which is placed on the wardrobe, he observes a red-haired girl with a scar on her lip playing with fire. They establish eye contact, but Aleksei's attention is then captured by a lamp that starts to turn on and off for no specific reason.

In the apartment, Ignat and his mother, Natalia, are present, and Natalia's belongings fall out of her bag. After that moment, two events of unusual and mysterious origin are depicted. Ignat helps his mother pick up the items but experiences an electric shock from an undefined object on the floor. Aleksei explains that he has a feeling that this moment has already happened, even though he is confused by the fact that it is his first time in that apartment. Ignat feels a déjà vu that resembles a reminiscence or a vague memory, often emotionally tinged. The next mystery is portrayed through a panning shot that initially shows an empty room. However, when the camera returns to the room, there is an elderly woman inside who asks Ignat to read a letter by Alexander Pushkin. Shortly after hearing the bell, Ignat opens the door to see his grandmother, but for some unknown reason, they do not recognize each other. Ignat returns to the room only to find that the elderly woman has disappeared.

Even though we never find out who the elderly woman in the room is, Tarkovsky describes her as "This is simply a woman who knits together the broken thread of time" (cited according to Skakov 2012, p. 121). It seems that Ignat has encountered a supernatural being that controls time. Like Aleksei, Ignat shows no emotions and is not disturbed by the unusual presence and disappearance of the elderly woman or her request for Aleksei to read. Additionally, during the scene, Tarkovsky deliberately breaks the rule of continuity editing to further create a sense of mystery.

Another child character in *Mirror* is Asafyev, an orphan from Leningrad. Tarkovsky shows him in a shot on a snowy hill by the river when a small bird lands on his head. Like Alexei, Asafyev is serious and deeply focused, and his gaze is directed to one point.

The bird is a crucial symbol here, associated in various spiritual traditions with transcendence, the soul, and flight to other worlds. It is also a Christian symbol of the Holy Spirit. Tarkovsky points out that a bird will never approach an evil being. Asafyev had behaved rudely in previous scenes, but to prevent the audience from thinking of him as a delinquent boy, Tarkovsky uses the motif of the bird to reveal Asafyev's true nature, which is that he is good, just like any child (Gianvito 2006, p. 72). The landing of the bird on Asafyev's head by the river can be compared with the description of Jesus' baptism in the river Jordan: "After Jesus was baptized, he came up from the water and behold, the heavens were opened, and he saw the Spirit of God descending like a dove coming upon him" (Matthew 3:16). In this sense, Asafyev could be likened to a Christ-like figure. However, the scene is interrupted by documentary footage of the fall of Berlin and the dropping of the atomic bombs. We return to Asafyev, who catches the bird with his hands,

but the narrative is once again interrupted by documentary footage, this time depicting the Cultural Revolution and the Soviet Union's disagreement with China over Zhenbao Island in 1969. His memories become premonitions of future hostilities, and a premonition or divination is a prophetic gift of clairvoyance. Asafyev is seen wearing a worn-out and torn coat with chapped lips, and his entire face and palms are frostbitten. His gloomy premonitions, appearance, and facial expressions reveal the significance of his name, "the one who is sad".

The transcendent reflection of Aleksei and Ignat is expressed through mysterious places where they encounter the unknown. Aleksei visits his neighbor, Mrs. Solovyova, whose house exudes a mysterious atmosphere. There, he encounters a red-haired girl portrayed by Tarkovsky as if she is within a mirror. However, we never find out if the mirror serves as a portal to another place or if Aleksei is merely fantasizing about the red-haired girl. Furthermore, Ignat, like his father, finds himself in a mysterious space where he encounters an elderly woman, whom Tarkovsky describes as having the ability to "knit together the broken thread of time". We can deduce that Ignat's sense of reminiscence in that scene is connected to Aleksei's observation of himself in the mirror and the fact that they are portrayed by the same actors. This also demonstrates how Ignat and Aleksei are inseparable from each other despite the passage of time. They are, in essence, the same person with a shared past, present, and future. While Ignat stands out as one of the few child characters not dressed in modest and worn-out clothing, Aleksei is depicted in very simple attire, and in some scenes, we see him walking barefoot through muddy forests. Transcendence is also reflected through the character of the boy Asafyev and his premonitions, i.e., the ability to foresee future events. Additionally, Tarkovsky's decision to portray a bird landing on Asafyev signifies the symbol of the Holy Spirit, representing the supernatural.

*4.4. Marta in Stalker ([Tarkovski 1979](#))*

At the heart of the narrative of this science fiction film is Stalker, whose occupation involves guiding people to a mysterious place where the usual laws of physics no longer apply–the Zone. It is assumed that a meteor or a flying object crashed there, but what makes the Zone significant is the Room, a place that grants the wishes of those who enter it. The entire Zone is off-limits to the public and under government control, making Stalker the only one able to approach it secretly. *Stalker* can be defined as a film about a metaphysical journey into a lost spiritual dimension ([Green 1993](#), p. 93), as well as a film about hope and a future that surpasses despair and death ([Robinson 2006](#), p. 465). Besides the religious symbolism in the journey itself and the enigmatic Zone, some authors consider Stalker as a Christ-like figure guiding us on a religious journey through challenges and temptations to reach the Zone where we will encounter God. He resembles a saint and martyr in appearance, while some call him a holy fool or "God's fool", and in some scenes, he even quotes the New Testament, specifically Christ's journey to Emmaus, where He meets disillusioned disciples and explains the necessity of His crucifixion ([Robinson 2006](#); [Chung 2016](#)).

When it comes to the child characters, one particularly striking scene is the last one in the film, where Marta is depicted as the "holy child". The scene is captured in a single take, beginning with a close-up of Marta's face and the camera slowly moving backward to achieve a medium shot. Initially, Marta is seen reading a book, and after putting it down, she recites Fyodor Tyutchev's poem I Love Your Eyes. Then, Marta demonstrates her supernatural ability of telekinesis, a hypnotic mental power to manipulate events and objects without physical contact. Using telekinesis, she moves three glasses on the table, which, according to Robinson, can carry multiple meanings. They could represent the three characters heading into the Zone, and the tall glass Marta knocks down may symbolize Stalker's fall "over the edge of faith, into unbelief"; his life marked by an apotheosis or his attainment of peace at the film's end. Alternatively, if we connect the three glasses to Christian themes, they represent the Holy Trinity ([Robinson 2006](#), p. 464). Marta is yet

another example of an unusually serious child character, her deep and contemplative gaze conveying that she knows something others do not. As the scene concludes, the camera slowly zooms back in to provide a close-up shot of Marta with her head resting on the table, looking sad and tired, as if burdened with the weight of the world, trembling due to the passing of a train. Tarkovsky, with this shot, once again expresses that gaze through a close-up.

Furthermore, there exists a visual connection between Marta and the mysterious Zone. Only the scenes set in the Zone and the frames featuring Marta are shot in color, while the rest of the film is captured in sepia tones. The apartment where Stalker's family resides gains color for the first time when we see Marta. Consequently, we can infer an inexplicable link between Marta and the Zone, suggesting that her supernatural abilities stem from her connection with the Zone, the sacred place. If the Zone is a sacred place shot in color, contrasting with all other scenes, then Marta is a sacred woman. The headscarf the young girl wears is thereby reminiscent of the veils often portrayed on the Catholic Church saints in sacred art, particularly the Virgin Mary.

Tarkovsky says about Marta: "She represents hope, quite simply. Children are always something hopeful. Probably because they are the future" (Gianvito 2006, p. 59). Tarkovsky describes Marta using the word "hope", and when we place this word in the context of the transcendent and the absolute, hope takes on the meaning of a source of religious beliefs. Turovskaya does not see this hope in Marta's telekinetic abilities or her spiritual development, which leads her to read Fyodor Tyutchev's poems. Instead, she sees hope in the simple act of uttering words that older individuals cannot comprehend (Turovskaya 1990, pp. 114–15).

Between Marta, Stalker, and the Zone, there exists a unique connection. Marta, a girl with supernatural abilities, the daughter of the Christ-like Stalker, who also possesses a supernatural gift, the "knowledge of the path" to the mysterious, unexplainable, and sacred place—the Zone, which could be seen as a sort of Paradise—a unity with the Absolute. Furthermore, Marta holds a mysterious connection with the Zone, the place from which her power emanates, and the attributes of which reflect God. As there is an evident connection between Marta, Stalker, and the Zone, we can infer that their relationship mirrors the Holy Trinity, where Marta represents the Holy Spirit. Her name carries the meaning of "mistress" and "servant", and the Holy Spirit, in Christian tradition, is the one who serves the faithful with an abundance of spiritual gifts.

### 4.5. Domenico's Son and Angela in Nostalghia (*Tarkovski 1983*)

*Nostalghia* is the sixth feature film by Andrei Tarkovsky and the first film he made during his exile in Italy. The movie follows the journey of a Russian writer, Andrei Gorchakov, during his visit to Italy to conduct research on the life of the Russian composer Pavel Sosnovsky. During this trip, he meets Domenico, who attempts to cross an old Roman pool while holding a lit candle, but as he cannot do it himself, he asks Gorchakov to do it for him. Thematically, the film focuses on the boundaries between cultures and the (in)ability of communication among people. These themes are shaped by Andrei's closed, repetitive, and circular dreams and memories, the repetition of monotonous sounds, and identical scenes.

In *Nostalghia*, two children appear: Domenico's son and Angela. Domenico's son, a five-year-old boy with blond hair and deep brown eyes whose name we never learn, appears in a scene shot in black and white. This scene depicts the police arriving at Domenico's house to take his family away. Tarkovsky uses parallel editing to show both the past and the present. The shots of Domenico walking through the place where he lost his family are interspersed with shots of the past, showing the same event. At one point, Domenico's son starts running, and Domenico chases after him as they push through a crowd of people. When Domenico catches up to his son, they both pause on the stairs. A close-up shot of the boy's face gains color, and the boy asks, "Is this the end of the world,

Dad?". This scene echoes what we have seen in previous films, portraying a serious child with a profound gaze captured in a close-up shot.

The reflection of transcendence in Domenico's son is evident primarily in the fact that he is Domenico's son, or in other words, the son of a holy fool who, like his father, gains enlightenment when he steps outside the house. This enlightenment is seen in the question he poses to his father with a very serious expression: "Is this the end of the world, Dad?" This question can be linked to the one that Jesus addresses to the Father on the cross: "My God, my God, why have you forsaken me?" (Mark 15:34). The boy's question expresses the pinnacle of his innocence, confusion, suffering, and the need for a paternal figure he can rely on to explain what is happening and why. Furthermore, we can relate the situation in which Domenico had placed his family to Plato's Allegory of the Cave. According to this allegory, three men are chained in a cave, only able to see what is in front of them. They merely observe shadows, which they consider to be real. One of the men is freed and goes outside, realizing there exists an entirely different world, but the sudden exposure to the light blinds him. Due to the fear of pain, the other two men refuse to leave the cave. In this context, Domenico can be associated with the enlightened man who refuses to let his family into the light, keeping them locked in the house for seven years. Moreover, the boy appears in Gorchakov's dreams, which take place in Russia. This illustrates a deep connection between Domenico and Gorchakov, both living estranged from their families, and the boy who seems almost bilocated, appearing in the dreams of a man who did not know him, akin to an angel or God's messenger. The boy's large, curious, and serious eyes, staring at the viewer and breaking the fourth wall, also evoke a sense of transcendence, as if questioning our conscience, inviting us to reflect and conveying something essential without words—almost on a metaphysical, telepathic level.

Angela is a mysterious little girl who climbs through the ruins of a sunken church and bears a symbolic name that explicitly alludes to a spiritual being—an angel. Her light brown hair is covered with a cap, and she wears a gray dress covered with a large shawl. For unknown reasons, Gorchakov enters the submerged church and encounters Angela there. We first see her peering through a hole in the wall, and Gorchakov meets her while delivering a monologue that ends with a question directed at her—is she satisfied with life? To which the girl gives a positive answer. This greatly delights Gorchakov, and it is the only moment in the film where we see him smiling, as if he senses a glimmer of faith and hope for a better future in the child's character. After Angela throws a pebble into the water, the recitation of the poem My Sight, My Strength, Dims. . . by Arseny Tarkovsky begins.

Initially, what makes Angela an expression of transcendence is her mysterious appearance and her name, which alludes to an angel—a spiritual being in Christianity and Judaism. The word is derived from the Greek language and denotes a messenger. Skakov believes that Angela is actually the third angel appearing in *Nostalghia*. The first angel is human-sized, standing near Gorchakov's house in Russia, and the second is a ceramic angel submerged at the entrance to the church (Skakov 2012, p. 183). P. Adams Sitney compares Angela with an angel depicted in the painting Presentation of Jesus in the Temple by a Renaissance painter, Vittore Carpaccio (Adams Sitney 2014, p. 237).

Domenico's son represents a deep connection between Gorchakov and Domenico, two holy fools; he serves as a kind of God's voice, reminding Gorchakov and Domenico that they must save the world. The boy's question reminds Domenico of the end of the world, and he appears in Gorčakov's dreams after accepting Domenico's request to cross the pool while holding a lit candle. In the end, both offer their lives as a sacrifice to save the world, just like Christ.

### 4.6. Gossen in The Sacrifice (*Tarkovski 1986*)

In the last feature-length film by Andrei Tarkovsky, we follow the story of a journalist and aesthetics professor named Alexander. He lives with his wife Adelaide, his stepdaughter Marta, and son Gossen in a house on the Swedish coast. Later, Alexander's friends join them as well. As fighter jets fly overhead, the news on the radio announces a general war

and the possibility of a nuclear holocaust. Panic grips the family, and Alexander makes a vow to God that he will renounce everyone he loves to prevent the catastrophe.

In his last film, Tarkovsky blurs the boundaries between dreams and reality, imagination and the material, and changes the meaning of visual and auditory codes. This creates a diegetic world that is both real and unreal. Throughout the film, he incorporates a series of Christian images and references. Alexander is portrayed as a saint or mystic, troubled by worries and going through the "dark night of the soul", and his journey is akin to that of a mythological hero. It starts with suffering and culminates in enlightenment (Robinson 2006, p. 498). Despite the perplexing dialogues, *The Sacrifice* reaches its conclusion: that love, hope, and faith in God can save humanity from self-destruction. Instead of attempting to dominate the natural world, man should return to a profound respect and harmony with it (Johnson and Petrie 1993, p. 183).

The boy, Gossen, plays a significant role in the first and last scenes of the film, where he is linked to the transcendent. In the opening scene, we witness Alexander and Gossen planting what Alexander refers to as a "barren and withered tree".

Alexander recounts the story of an old Orthodox monk named Pamve who once planted a barren tree on a mountain and then commanded the young monk Ioann Kolov to water it every day until it came back to life. The young monk faithfully carried out this task for three years until the infertile tree bore fruit. This act makes Alexander think, and he also believes that the ritual repetition of the same act every day at the same hour could change something in the world. Tarkovsky refers to this act of planting as an act of faith (Green 1993, p. 124). Additionally, Alexander notices that Gossen attempts to utter the words of John's Gospel, "In the beginning was the Word" (John 1,1), but he struggles to articulate the words coherently.

In the final scene, we see Gossen taking on his father, Alexander's, role and watering the barren tree, following the example of the young monk from the story at the beginning of the film. Gossen lies beneath the tree and finally speaks after his operation, quoting the words from the Gospel of John: "In the beginning was the Word". He then asks, "Why is it like that, Dad?" Gossen, like Domenico's son in *Nostalghia*, poses a deep and religious question, but this time, it is not shrouded in fear and suffering; instead, it is driven by curiosity and positivity. The question is directed at his father, Alexander, but it can also be interpreted metaphorically as if it were addressed to God. Just as God sent His only Son to grant eternal life to all who believe in Him, Alexander has passed on to his son Gossen the responsibility of caring for the Tree of Life, which is meant to save the world. By taking care of the tree, Gossen, in a way, becomes a monk. After a single watering, the change in the world that Aleksandr spoke of is already occurring: Gossen has spoken, and his first words are the sacred words from the Gospel of John. This reflects a new beginning in which the Tree of Life will bear fruit again, specifically, the knowledge between good and evil that man lost when he ate the forbidden fruit, according to the Book of Genesis, but also when initiating a global war. The scene is further elevated by Bach's aria from the St. Matthew Passion. The camera slowly moves up toward the top of the tree, marking both the end of the film and Tarkovsky's entire filmography.

Johnson and Petrie interpret the ending as an affirmation of the potential that man can achieve when driven by love and faith. This affirmation is expressed through Bach's music and the dead tree that Gossen, with his careful care, will bring back to life (Johnson and Petrie 1993, pp. 183–84). Turovskaya argues that the attempt at resurrection is the guiding idea behind every film by Andrei Tarkovsky, and it is finally realized in *The Sacrifice* (Turovskaya 1990, p. 149).

Gossen is portrayed somewhat differently from children in Tarkovsky's previous films. He is not dressed in worn-out and poor clothing, and the hat he wears in most scenes obscures his face. Additionally, Tarkovsky never uses close-ups of Gossen; instead, he employs medium shots, medium–long shots, and long shots. By choosing to depict Gossen solely through these specific film shots, Tarkovsky ensures that Gossen's presence is never

dominant in the scene, and he does not draw attention to himself directly. However, we are aware of his subtle presence, transforming him into a silent observer.

The tree that Alexander and Gossen plant and that comes to life can be connected to the Tree of Life, a symbol of the life struggle and immortality found in various mythologies. It is the tree that grants knowledge of good and evil or the tree that God caused to flourish. The Tree of Life appears in the Book of Genesis in the Old Testament. In Christianity, the Tree of Life becomes associated with the cross, which replaces the Tree of Knowledge of good and evil from the Old Testament in the earthly paradise, and in the Middle Ages, the cross sometimes takes the form of a tree. In the film's prelude, Tarkovsky includes details from Leonardo da Vinci's Adoration of the Magi painting, depicting the Tree of Life. This painting is later seen again when Alexander presents it to Gossen while he is sleeping. Additionally, Gossen's pose, similar to the reclining Buddha, refers to a statue or image representing Buddha during his last days as he enters Parinirvana, the stage of great salvation after death that can be attained only by enlightened souls. As God, through Jesus Christ, initiated a new and extraordinary reality, Alexander, through Gossen, begins a new reality that will set humanity on the right path. Gossen becomes a sign of Alexander's entry into this world, demonstrated by Gossen continuing their "sacred ritual" of watering the tree. Thus, a parallel can be drawn between Alexander and Gossen with God the Father and Jesus Christ, the Son.

## 5. Conclusions

Andrei Tarkovsky is one of those film auteurs whose work will always remain relevant due to his serious and profound approach to filmmaking, the human spirit, and transcendent questions. His deep religious beliefs are strongly present in each of his films. His movies are rich in Christian motifs, so it is not surprising that Tarkovsky emphasizes the importance of portraying and manifesting God in art, including (his own) filmmaking, by comparing his films to prayers that bring man closer to God. At the heart of Tarkovsky's prayer is the image of a child, which draws the viewer closer to the transcendent, or in other words, to God.

To support the initial thesis that Tarkovsky uses child characters to express the transcendent, this paper analyzes nine child characters from six feature-length films: seven boys (Ivan Bondarev, Boriska, Aleksei, Ignat, Asafyev, Domenico's son, and Gossen) and two girls (Angela and Marta). Throughout the film analysis, certain recurring characteristics of these children were noticed, which, in our opinion, point to the transcendent. These child characters possess a deep and penetrating gaze that creates the impression of telepathic communication. This gaze is often depicted in close-up shots, allowing the viewers to see the faces up close and almost immerse themselves in them, thus entering a different world of the sacred and supernatural. The expressions on the children's faces are typically solemn, serious, and somewhat sad and worried, as they are surrounded by suffering and uncertainty, both due to their difficult family situations and the global suffering caused by wars.

Children in Tarkovsky's films pose deep and ultimately religious questions directed both at their earthly fathers and, on a higher level, at God the Father: "Why is it like that, Dad?" and "Is this the end of the world, Dad?" These questions are of a theodical nature, essentially asking about the origin of evil in the world; whether evil will have the last word; and if there is a God, why does He allow the innocent, especially children, to suffer? Tarkovsky does not provide clear answers; instead, he, together with the children, contemplates these questions. However, the concluding scenes do not convey pessimism but rather hope that goodness will triumph, and in Tarkovsky's films, goodness is embodied in the characters of children who hold the answer.

In Tarkovsky's films, children share certain similarities with Christ-like figures in certain aspects. One notable similarity is that, except for Gossen, all the children are dressed in worn-out and modest clothing, and sometimes they are barefoot, in line with the Christian notion of God who "emptied" himself and did not come into the world in

splendor like a king; rather, he lived on Earth in humble attire. Furthermore, they perform miracles, i.e., possess supernatural powers, which also connects them to Jesus Christ: Ivan Bondarev levitates, Boriska intuitively knows the secret of bell casting, guided by the supernatural; Ignat experiences reminiscences, Asafyev has premonitions, Marta moves objects using telekinesis, and Gossen revives the Tree of Life. Many of these characters are orphans, or we have no knowledge about one or both of their parents, such as Ivan, Boriska, Angela, and Asafyev, which alludes to Jesus, who, according to Christian belief, was incarnated by the Holy Spirit in the womb of Mary and, in that sense, did not have a biological father.

When it comes to the psychological characteristics of children in Tarkovsky's films, we have noticed that they share common traits: they are introverted, anxious, and lonely beings resembling hermits who spend little time in the company of others. Tarkovsky rarely shows children in the company of others, and we almost never see them interacting with peers. Additionally, we rarely see them smiling or playful, and they do not leave an impression of happy and carefree children as often depicted in films, especially in Hollywood cinematography. There is not a single portrayal of a child playing, which means Tarkovsky deprives them of one of the fundamental childlike traits. With their demeanor, they give the impression of "carrying the weight of the world", and we can observe recurring behavioral characteristics. They are often seen silently observing their surroundings with deep gazes. Most of the children in Andrei Tarkovsky's films move slowly and gently, never drawing attention to themselves, except for Ivan Bondarev in *Ivan's Childhood* and Boriska in *Andrei Rublev*.

In all of Andrei Tarkovsky's films, children are supporting characters, except in *Ivan's Childhood*. Despite their lesser presence in the plot, children have a powerful impact on other characters: Ivan Bondarev's disappearance motivates Galtsev to spend the entire war searching for him, and his death deeply affects Galtsev in the end; Andrei Rublev breaks his sixteen-year vow of silence and resumes his artistic creation after witnessing the bell made by Boriska; Aleksei and Domenico deeply care for their sons; Gorchakov experiences happiness for the first time after meeting Angela. Additionally, Domenico's son and Gossen are the descendants of holy fools who, through their behavior, show that one day they themselves could become "God's fools": Christ-like characters with supernatural abilities.

If we know that Ivan Bondarev exists between reality and dreams, Boriska's construction site resembles a sacred place of religious rituals, Ignat comes into contact with a supernatural being in the form of the elderly woman, Marta is connected to the inexplicable Zone, Angela spends time in a sunken church, and Gossen revives the Tree of Life on the shore, we can conclude that all the children are placed in mysterious environments that are atypical for their age and prone to various interpretations. In these environments, they have their feet on the ground but their heads touching the heavenly. When discussing the filming style of child characters, the close-up shot becomes a key tool for expressing their transcendence because this is exactly how Tarkovsky manages to capture a deep and penetrating gaze that leaves the viewer with the impression of telepathic communication and thus manages to get extremely close to him. The close-up shot is a common technique used by other filmmakers as well to communicate the transcendent, especially Carl T. Dreyer and Krzysztof Kieślowski. Only Angela and Gossen are never depicted in close-up shots but rather in medium, medium–long, and long shots. The explanation for these stylistic choices can be found in the fact that Angela represents a guardian angel who watches over her ward from a certain distance, while Gossen is a silent and distant observer of events. The other seven characters always get a moment when they are shown in close-up shots, with which Tarkovsky further emphasizes the transcendent.

In summary, Tarkovsky uses child characters to express the transcendent by giving them defined recurring characteristics: a deep and penetrating gaze, supernatural powers, mysterious surroundings, a strong influence on other characters, posing religious questions, and a sense of solitude resembling hermits, and sometimes an allusion to Christ-like figures. He often films them in close-up shots and dresses them in modest and poor clothing. It is

interesting to note that Tarkovsky ends his last film in the same way he began his first, with a child in the foreground, demonstrating how important depicting children and childhood was to him. Thus, the image of the child becomes the beginning and the end, the alpha and omega of Andrei Tarkovsky's entire filmography. In an atmosphere of uncertain future, fear of what lies ahead, and man's destructive capacity for self-destruction through war and hostilities that Tarkovsky philosophically and theologically contemplates through his films, and which remain relevant even today, children represent the only hope for a better future. They are innocent, unspoiled, intuitive, and therefore in touch with the transcendent and a reflection of transcendence itself.

**Author Contributions:** Conceptualization, M.P. and I.S.G.; methodology, M.P. and I.S.G.; resources, M.P. and I.S.G.; writing—original draft preparation, M.P.; writing—review and editing, I.S.G. All authors have read and agreed to the published version of the manuscript.

**Funding:** This research received no external funding.

**Conflicts of Interest:** The authors declare no conflict of interest.

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
