# Peer review of "Children as a Reflection of Transcendence in the Filmography of Andrei Tarkovsky"

_religions, doi:10.3390/rel14091138_

Round 1

Reviewer 1 Report

This is a very interesting paper. To be honest, I am not a film critic, but a theologian and educator, so I read this article from a theological-educational perspective, and I rate this text highly from both perspectives. This is in line with the well-known belief of Maria Montessori that peace comes to the world thanks to children, which is also mentioned in the inscription on her grave. It is located in the small coastal town of Noordwijk in the Netherlands, where Dr. Montessori died. Her tombstone reads: "I beg the dear, all-powerful children to join me in creating peace in man and in the world."

The power of a child combined with the creative strength of an artist may form a real acting force in the world that works to change the social environment according to the principles of peace and love.

Tarkovsky was both a theologian and an artist, and even maybe a child with its sense of wonder when exploring the surrounding environment. Such a strong connection between art and faith is not so common nowadays and it’s almost impossible to defend now Tarkovsky’s conviction: “I don’t believe in art without God”. But on the other side, this reveals an important truth related to faith transmission that it becomes fruitless if deprived of art (pure conceptual education) or when overloaded by entertainment (pure amusement and well-being).

Recalling this important feature of the artistic profession, which is being a "detector of the absolute" in a fragile and unstable world, is the strength of the article, although highly debatable from the perspective of contemporary art theory or psychology of creativity.

In terms of methodology and structure, the work, in my opinion, does not require any corrections. It has a good introduction, research questions and methodology are clearly presented and then summarized in a final discussion. The only thing missing is a reference to critics who could shed new light on Tarkovsky's film work or stylistic devices employed it the movies. Nevertheless, it is astonishing that such a valuable production took place in a communist country.

Finally, I’d like to notice two little mistakes:

Line 132 – Ivan Pavalo – should be: John Paul (as in line 138)

Lines 851-852 – Christians believe Jesus had a biological mother but no biological father. This is a dogma of the early Church. The Council of Ephesus in 431 declared Mary to be the Theotokos because her Son Jesus is both God and man: one divine person of two natures (divine and human) intimately and hypostatically united. Therefore, this passage should be rephrased in order not to make a mistake.

Author Response

Point 1: Line 132 – Ivan Pavalo – should be: John Paul (as in line 138)

Response 1: On the page 3, we changed Ivan Pavalo in John Paul.

Point 2: Lines 851-852 – Christians believe Jesus had a biological mother but no biological father. This is a dogma of the early Church. The Council of Ephesus in 431 declared Mary to be the Theotokos because her Son Jesus is both God and man: one divine person of two natures (divine and human) intimately and hypostatically united. Therefore, this passage should be rephrased in order not to make a mistake.

Response 2: In the lines 918-922 we changed the meaning of the sentence regarding to the Jesus’s conception to avoid the theological misunderstanding:

"Many of these characters are orphans or we have no knowledge about one or both of their parents, such as Ivan, Boriska, Angela, and Asafyev, which alludes to Jesus, who, according to Christian belief, was incarnated by the Holy Spirit in the womb of Mary and, in that sense, did not have biological father."

Reviewer 2 Report

The topic of the article is extremely interesting and topical both because of the increasingly observed gap between art and transcendent values and the encouragement of Christian churches to use art as a 'via pulcherinibus' leading to God. The article is structured correctly, clearly, and consistent. Only the last subsection called 'discussion' is questionable, in which there are no issues for possible discussion, rather conclusions arising from the analysis of the selected materials - perhaps the title 'conclusions' would be more appropriate.

One can see a certain unevenness in the reference to sources when discussing individual films, but this may be related to the fact that some are more extensively developed, others less so.

In lines 94-97 and 149-150, the same thought is unnecessarily repeated.

Apart from these minor stumbling blocks, the authors of the article are to be congratulated for their diligent workmanship and an important contribution to the rapprochement between theology and culture that is so much needed given the increasingly observed separation of these two paths towards transcendence or God.

Author Response

Point 1: Only the last subsection called 'discussion' is questionable, in which there are no issues for possible discussion, rather conclusions arising from the analysis of the selected materials - perhaps the title 'conclusions' would be more appropriate.

Response 1: On the page 9, chapter 5, we renamed Discussion into Conclusion.

Point 2: In lines 94-97 and 149-150, the same thought is unnecessarily repeated.

Response 2:  On the page 3, line 151-153, we erased a part of the sentence which was indicated as a duplicate: „…the style itself is not inherently transcendental or religious; rather, it serves as a means of approaching the transcendent. According to Schrader“.

Reviewer 3 Report

This article would make an interesting contribution to the journal. It is well written, clearly set out and argues coherently for the transcendent significance of children in Tarkovsky's films. I would strongly recommend it for publication.

The one area for improvement I noted was in the contextualisation of the use of children and the child's eye view in films. The author contrasts Tarkovsky with films that sentimentality children, but there's a rich seam of films that are perhaps closer to Tarkovsky's appproach including those where a non-adult and otherworldly element, or theodical questions are introduced. One could think of Laughton's The Night of the Hunter, Erice's The Spirit of the Beehive, Malik's The Tree of Life, for example, and the author could no doubt think of others. More significant for Tarkovsky is the portrayal of children in other Soviet Thaw films. There is a short article on this by Alena Ianushko in the Global Journal of Social Science Vol 21, Issue 5, 2021, 'The Image of a Child in the Cinema of the Soviet Thaw', which is not developed at length but is worth a ment

It would be good to have some citations from more recent publications.
